# Insights from Mass Spectrometry-Based Proteomics on *Cryptococcus neoformans*

**DOI:** 10.3390/jof11070529

**Published:** 2025-07-17

**Authors:** Jovany Jordan Betancourt, Kirsten Nielsen

**Affiliations:** 1Department of Microbiology and Immunology, University of Minnesota, Minneapolis, MN 55455, USA; betan041@umn.edu; 2Department of Biomedical Sciences and Pathobiology, Virginia Tech University, Blacksburg, VA 24060, USA

**Keywords:** *Cryptococcus neoformans*, cryptococcosis, proteomics, immune response, host–pathogen interactions

## Abstract

*Cryptococcus neoformans* is an opportunistic fungal pathogen and causative agent of cryptococcosis and cryptococcal meningitis (CM). Cryptococcal disease accounts for up to 19% of AIDS-related mortalities globally, warranting its label as a pathogen of critical priority by the World Health Organization. Standard treatments for CM rely heavily on high doses of antifungal agents for long periods of time, contributing to the growing issue of antifungal resistance. Moreover, mortality rates for CM are still incredibly high (13–78%). Attempts to create new and effective treatments have been slow due to the complex and diverse set of immune-evasive and survival-enhancing virulence factors that *C. neoformans* employs. To bolster the development of better clinical tools, deeper study into host–Cryptococcus proteomes is needed to identify clinically relevant proteins, pathways, antigens, and beneficial host response mechanisms. Mass spectrometry-based proteomics approaches serve as invaluable tools for investigating these complex questions. Here, we discuss some of the insights into cryptococcal disease and biology learned using proteomics, including target proteins and pathways regulating Cryptococcus virulence factors, metabolism, and host defense responses. By utilizing proteomics to probe deeper into these protein interaction networks, new clinical tools for detecting, diagnosing, and treating *C. neoformans* can be developed.

## 1. Introduction

*Cryptococcus neoformans* is a fungal pathogen that causes substantial mortality and morbidity in immunocompromised individuals. Normally found in soil, decaying plant material, or bird feces, human *C. neoformans* infection begins after inhalation of aerosolized spores or fungal cells which infect the lower respiratory tract and alveoli. During initial infection in immunocompetent individuals, tissue-resident and innate immune cells clear or contain the infection, followed by the establishment of adaptive memory responses that respond to subsequent exposures [1]. In urban environments, 70% of individuals >5 years old were serologically positive for anti-Cryptococcus antibodies, providing evidence of exposure to *C. neoformans* early in life [2]. While cryptococcal infection starts in the lungs, it is the fungus’s ability to disseminate to the central nervous system following immunosuppression that is of greatest concern. By far the largest risk factor for disseminated cryptococcal disease is the depletion of CD4+ T helper cells (<200 cells/μL) after HIV infection [3,4]. The estimated global incidence of cryptococcal disease, or cryptococcosis, in HIV-infected individuals in 2020 was 179,000 cases and accounted for approximately 19% of AIDS-related mortality [3]. Indeed, *C. neoformans* is a pathogen of global concern, with the World Health Organization labeling it as a pathogen of critical priority [5].

Following immunosuppression, *C. neoformans* escapes the pulmonary system to disseminate to other organs including the brain, which causes the worst manifestation of cryptococcal disease, cryptococcal meningitis (CM) [6,7,8]. Symptoms of CM include fever, severe headache, confusion, and elevated intracranial pressure with mortality rates ranging from 13 to 78% [9,10,11]. To improve patient outcomes, early detection of cryptococcal dissemination and initiation of antifungals are crucial. The development of Cryptococcus-specific antigen lateral flow assay strips (CrAg LFAs) by IMMY has enhanced our ability to detect *C. neoformans* dissemination and has become a “gold standard” in fungal diagnostics [11,12,13]. However, cases of seronegative CM that elude CrAg LFA detection underscore the need for better diagnostics for detecting low-burden infections [14,15].

The standard treatment regimen for cryptococcosis and CM involves three phases of antifungal treatments: an initial two-week induction phase with several high-dose antifungals, followed by a consolidation phase of at least eight weeks of mid-dose antifungals and initiation of antiretroviral therapy, and a long-term maintenance phase with over a year of lower-dose antifungals [16]. The reliance on several rounds of antifungal therapies for long periods of time has contributed to the growing threat of antifungal-resistant *C. neoformans* [17,18,19]. Immunomodulation has been suggested as an alternative mechanism to treat CM. Cryptococcal infection is known to induce nonprotective type-2 immune responses while suppressing the more beneficial type-1 responses [20,21,22]. However, clinical trials investigating the use of adjunctive interferon γ, the prototypical type-1 cytokine, to treat HIV-associated CM showed no benefit on mortality rates, despite reductions in fungal burden [23]. The lack of new and effective antifungal and immunomodulatory therapies for CM and cryptococcosis highlights the need for better models of cryptococcosis and more research into host–Cryptococcus interactions.

*C. neoformans* produces several virulence factors that promote its survival during infection [24,25]. Its characteristic polysaccharide capsule shields the fungus from phagocytosis and chemical damage [26]. Changes in cell wall organization and chitin composition modulate the recognition of immune cells [21,27,28]. Cryptococcus melanin production enhances survivability in harsh environments and against free radicals [29,30,31]. Its secreted extracellular vesicles modulate immune cell behavior and transport virulence factor precursors [32,33]. Cryptococcus also forms titan cells, which are large polyploid cells, 15–100 µm in diameter, that produce daughter cells that are highly resistant to fungicidal mechanisms [19,34,35,36]. Understanding these virulence factors, and the enzymes and pathways that produce them, is important for the development of clinical tools. More research into immune escape strategies, regulatory mechanisms, metabolism, signaling, protein production, and protein interaction networks in relation to known and as yet unknown virulence factors is needed to identify targets for new clinical therapeutics.

The use of mass spectrometry-based proteomics for studying pathogens and host–pathogen interactions has increased over the last two decades [37,38,39,40]. These analytical tools can be used to quantify protein abundance, determine protein pathway enrichment, identify protein modifications, and study protein interactions under various experimental conditions. Additionally, proteomics can be used to evaluate the efficacy of novel therapeutics including anti-infectives and vaccines in vitro and in vivo. To this effect, proteomics analysis of *C. neoformans* has become a valuable method for experimentally evaluating changes in protein behavior during stress response, infection, and genetic manipulation, as well as probing the host responses to cryptococcal infection [41,42,43]. In this review, we will explore some of the insights into *C. neoformans* biology and pathogenesis gleaned from proteomics analysis over the years. By understanding the benefits of utilizing proteomics to study *C. neoformans*, we can build upon the methodologies and experimental designs described here to aid in the development of new and effective clinical interventions to improve cryptococcal disease outcomes worldwide.

## 2. Approaches to *C. neoformans* Proteomics Analysis

Mass spectrometry-based proteomics techniques have undergone much advancement over the last two decades. Improvements in mass spectrometry hardware, sample preparation methods, and data analysis software have expanded the ability to detect and identify a wider range of proteins and peptides from complex samples. Still, careful consideration of protocol workflows is needed prior to the start of an experiment to maximize the detection of target proteins and minimize sample loss.

The general proteomics pipeline involves digesting proteins into peptides, ionizing them, detecting peptides using mass–charge ratios, and bioinformatically identifying the proteins in a sample. Variations in experimental setup have been utilized in *C. neoformans* research to optimize the detection of specific host or fungal proteins, maintain protein structure or post-translational modifications (PTMs), quantify protein abundances, or characterize phenotypes. For global proteomics analysis, shotgun proteomics is employed for an unguided sampling of all proteins present in a sample [44,45]. Comparative proteomics methods are used to analyze proteomic changes caused by experimental or clinical manipulations, or to make comparisons between different organisms’ proteomes [46,47,48,49]. Subtractive proteomics is useful for analyzing infected tissues and focusing on unique Cryptococcus-specific proteins by removing the host proteome [50,51]. To quantify the abundance or change in abundance of host or fungal proteins, labeled (TMT, iTRAQ, SILAC) or unlabeled (label-free quantification) approaches can be used [52,53]. Proteomics can also be employed to study protein PTMs, such as phosphorylation [42,43,54,55,56] and acetylation [46,57], which have also been shown to be significant in modulating protein activity and function. Finally, proteomics can be combined with other omics methods to relate protein presence and abundance to transcriptional expression (proteotranscriptomics) [58,59] or genetic mutations (proteogenomics) [42,59,60,61] and provide deeper insight into the complex mechanisms that control host–Cryptococcus interactions.

Sample preparation techniques have evolved to improve protein detection and minimize sample loss. However, processing *C. neoformans* poses unique challenges due to the hardiness of the cell wall and polysaccharide capsule that do not exist for typical mammalian tissue samples. To ensure complete lysis of the fungal cell and protein extraction, a combination of physical and chemical lysis techniques with detergents may need to be employed simultaneously. The Geddes-McAlister group at the University of Guelph have published several methodology papers and protocols that detail *C. neoformans* sample preparation techniques for mass spectrometry-based proteomics and phosphoproteomics, including a video protocol for label-free quantitative proteomics [52,54,62]. These resources are incredibly helpful for standardizing sample preparation and proteomics workflows, as small changes to sample preparation can have significant impacts on downstream protein identification.

Finally, several databases exist that can be useful for downloading previously generated proteomics data and storing new proteomes. UniProt, FungiDB [63], and the European Molecular Biology Laboratory (EMBL) Proteomics Identifications Database (PRIDE) [64] are large repositories of reference and experimental proteomes. On UniProt, the main reference proteome for *C. neoformans* is the *Cryptococcus deneoformans* serotype D (strain JEC21/ATCC MYA-565) entry (Taxon ID 214684, Proteome ID UP000002149, 6740 proteins, Table 1). Many research groups use the serotype A H99 or KN99α as their preferred reference isolate, however, and may come across discrepancies between the serotype D *C. deneoformans* reference proteome and the serotype A *C. neoformans* strains. UniProt possesses a *Cryptococcus neoformans* serotype A (strain H99/ATCC 208821/CBS 10515/FGSC 9487) entry (Taxon ID 235443, Proteome ID UP000010091, 7429 proteins, Table 1), which may better align with commonly used experimental references. While *C. deneoformans* and *C. neoformans* have a mean nucleotide identity of approximately 86% [65], there is a substantial difference in protein identifications depending on the proteome reference used (Figure 1). Examples of these annotation difficulties are shown in Figure 1 in which we analyzed peptides from a serotype A *C. neoformans* clinical isolate and conducted peptide searches with DIA-NN (Ver. 1.9.2, Ralser Group, University of Cambridge, Cambridge, UK [66]) using either the reference *C. deneoformans* or *C. neoformans* proteomes. We identified 2813 proteins using the reference *C. deneoformans* proteome but 3349 proteins using the *C. neoformans* proteome. Only 1690 proteins (37.7%) overlapped between both searches. Importantly, proteome entries in UniProt are predicted from submitted genomes and then annotated afterwards to confirm the protein entries. The JEC21 proteome is defined as the *C. neoformans* “reference proteome” on UniProt and is better annotated than the other proteomes. Thus, the observed differences between the Cryptococcus proteomes likely underscore the importance of creating high-quality and well-annotated reference proteomes for the various Cryptococcus species.

Additionally, databases of Cryptococcus proteins related to antibody and immune epitopes (Immune Epitope Database and Tools) [67], extracellular vesicles (ExVe) [68], transcription factors (*Cryptococcus neoformans* TF Phenome Database) [69], and co-functional protein and gene networks (CryptoNet) [70] are good resources for validating the results of proteomics studies. A database of Matrix-Assisted Laser Desorption/Ionization Time-of-Flight (MALDI-TOF) mass spectrometry spectra to clinically differentiate between *C. neoformans* and *Cryptococcus gattii* was created by Bernhard et al. (CryptoType) [71]. Researchers should make full use of and contribute to these databases to promote data sharing, make direct comparisons between studies, and allow for the retrospective analysis of previous proteomics studies using new tools.

## 3. Proteomics of the *C. neoformans* Organism

Mass spectrometry-based proteomics is an effective tool for investigating *C. neoformans* and has led to numerous discoveries of proteins and pathways involved in the virulence, survival, and metabolism of this fungal organism. Here, we will discuss some of the characteristics of Cryptococcus and the advancements made using proteomics.

### 3.1. Capsule and Cell Wall

*C. neoformans* encodes several virulence factors that aid its survival in harsh conditions, such as inside an infected host. The polysaccharide capsule is a key virulence factor and prevents fungal killing by physically interfering with phagocytosis, disrupting immune responses, and protecting *C. neoformans* from the contents of the lysosome [72,73,74,75]. The capsule is primarily composed of glucuronoxylomannan, galactoglucoxylomannan, and immunomodulatory mannoproteins [75,76]. The cAMP/protein kinase A (PKA) signaling pathway is important for the production of several virulence factors including capsule [77]. To identify the pathways involved in PKA-dependent virulence factor production, Geddes et al. utilized quantitative proteomics to determine changes in pathway enrichment between wild-type and PKA-deficient *C. neoformans* strains [26]. They found that the ubiquitin–proteasome pathway (UPP) was enriched in the PKA-complete strain and that selectively targeting UPP with the FDA-approved proteasome inhibitor bortezomib negatively impacted capsule production, providing support for using proteostasis agents to treat *C. neoformans*.

The cell wall of *C. neoformans* is also of interest due to its role in facilitating attachment of the capsule in addition to containing immunoreactive structural molecules such as chitin, chitosan, glucan, and the virulence factor melanin [24,27,28,29,30,78]. Melanin is a pigment produced by a variety of organisms to protect from environmental hazards such as radiation [31]. In *C. neoformans*, melanin is an important structural and defense component that prevents damage from reactive oxygen species present in lysosomes and temperature stress from host conditions, and modulates cytokine profiles to impair fungicidal immune responses [29,30,31,78]. To investigate melanin trafficking to the cell wall, Camacho et al. demonstrated using proteomics that four virulence-related *C. neoformans* proteins, Cig1, Blp1, Qsp1, and CNI3590 macrophage-activating glycoprotein, were associated with melanin granules containing the building blocks of the cell wall. These findings suggest a connection between melanin production, cell wall synthesis, and the secretion of immunomodulatory virulence factors that are dependent on related pathways. Indeed, in addition to capsule synthesis, PKA signaling is necessary for melanin production as well [26,77]. Thus, these studies showcase how proteomics can be used to identify how proteins of different virulence “mechanisms” may travel together in previously unknown ways.

### 3.2. Extracellular Vesicles and Secretome

Virulence factors may also be secreted from the *C. neoformans* cell inside extracellular vesicles (EVs) to influence its surroundings. Understanding the composition and formation of EVs is important for knowing the mechanisms of indirect activity that *C. neoformans* exerts on local host tissue and other fungal cells. Cryptococcus EVs can contain a variety of effector molecules including signaling proteins, metalloproteases, capsule and cell wall components, and virulence factor precursors [32,33,68,79]. Proteomics approaches have revealed a number of EV-associated proteins in several fungal pathogens [79]. Rodrigues et al. showed that *C. neoformans* EVs contained 76 proteins, many of which were related to pathogenesis and virulence including capsule components, heat shock proteins, and virulence factor-synthesizing enzymes [33]. They described *C. neoformans* EVs as “virulence factor delivery bags”, emphasizing their importance in transporting immunomodulatory molecules to host immune cells. To determine the identity of acid phosphatases that were seen to be secreted by patient-derived *C. neoformans* isolates, Lev et al. used proteomics and found that Aph1 was not only the main acid phosphatase secreted by *C. neoformans*, but that deleting it caused *C. neoformans* to be significantly less virulent in both murine and *Galleria* in vivo models [80]. Another family of proteins that *C. neoformans* secretes are peptidases which cleave proteins and peptides and can facilitate movement through extracellular matrices. The *C. neoformans* peptidase secretome was elucidated by Clarke et al. who found that growth conditions on YPD versus DMEM impacted peptidase expression [81]. Moreover, Clarke et al. identified a previously uncharacterized peptidase, named May1 (major aspartyl peptidase 1), which exhibited significant proteolytic activity and was necessary for infection virulence. Finally, the secretion of cellular cargo is dependent on transportation through endoplasmic reticulum (ER) channels. Santiago-Tirado et al. utilized proteomics to show that the ER channel protein Sbh1 was important for the effective transport of secretory and transmembrane proteins in conditions mimicking the host (37 °C and 5% CO_2_) [82]. When Sbh1 was deleted, the *C. neoformans* strain became avirulent in mice, underscoring the significance of secretory pathways in fungal survival during cryptococcal infection.

### 3.3. Host Envrionmental Stress Response

Unveiling the mechanisms that facilitate *C. neoformans* survival in host conditions can open the door to new antifungal therapeutics that counter these defense strategies. Proteomics analysis is becoming an important tool for studying these adaptation mechanisms and identifying candidate proteins and pathways that can be exploited to hamper *C. neoformans*’s ability to infect a host.

*C. neoformans* undergoes significant phenotypic changes when under host-like or harsh conditions to improve survivability. One of the first environmental changes the fungus must respond to is the much warmer host temperature of 37 °C. Calcineurin is a calcium-sensing phosphatase activated by calmodulin that dephosphorylates various targets in response to increases in Ca^2+^ levels caused by heat stress [83]. Calcineurin is required for *C. neoformans* to grow at host-like temperatures, as shown by *C. neoformans*’s susceptibility to heat stress following genetic knockout or inhibition of Ca^2+^ sensing with FK506 [84]. To elucidate targets of calcineurin dephosphorylation that are implicated in fungal survival, Kozubowski et al. and Park et al. both utilized quantitative proteomics to identify substrates associated with calcineurin activity [83,85]. Kozubowski et al. used an immunoprecipitation strategy to capture proteins bound to calcineurin before proteomics analysis and found that a variety of proteins associated with calcineurin, including the vesicular coating complexes COPI and COPII [85]. Park et al. utilized a quantitative phosphoproteomics approach to compare phosphopeptide profiles between calcineurin-activated and -inactivated conditions [83]. Their findings confirmed that calcineurin acts transcriptionally with the zinc finger transcription factor Crz1 and found novel post-transcriptional targets: RNA-binding proteins Lhp1, Puf4, and Pbp1. Fungal Ca^2+^-sensing pathways have long been seen as attractive targets for clinical therapies; however, the overlap between human and fungal pathways and immunosuppressive effects precludes using FK506, also known as the FDA-approved drug tacrolimus, as an antifungal therapy. Instead, these proteomics studies reveal possible alternative candidates by targeting the downstream substrates of calcineurin activity.

In addition to calcineurin, heat shock protein 90 (Hsp90) is another key protein involved in the phenotypic changes that promote survival in host conditions [86,87]. Hsp90 is a chaperone protein that prevents the degradation of bound proteins, extending their lifespan and functionality. Ball et al. found in their proteomics analysis of *C. neoformans* in zinc-limited conditions that Wos2, a co-chaperone of Hsp90, was involved in promoting thermal tolerance and virulence factor production during zinc excess [88]. Following this discovery, Ball et al. performed high-resolution quantitative proteomics evaluating the role of Wos2 in stress-response phenotypes and cryptococcal infection [89]. They found that Wos2 deletion significantly altered *C. neoformans*’s ability to tolerate high temperatures, high salinity, oxidative stress, and antifungal treatments, and exhibited reduced virulence.

Septins are cytoskeletal GTPases that are required for fungal survival and growth at host-like temperatures [90,91,92]. Deletion of the septins Cdc3 and Cdc12 in *C. neoformans* significantly hinders its ability to form hyphae and infect *Galleria mellonella* [91]. To determine the mechanisms by which Cdc3 and Cdc12 promote *C. neoformans* survival at 37 °C, Martinez et al. used tandem mass spectrometry-based proteomics to identify septin-associated proteins [92]. Interestingly, they found that heat stress led to a significant decrease in septin abundance, in contrary to their expectations that growth in host-like conditions would drive their expression. However, Martinez et al. showed that septins associated with two key proteins at 37 °C: the actin-stabilizing profilin Pfn and the antifungal resistance ABC transporter Afr1. These protein associations were confirmed with immunoprecipitation showing that septins facilitate *C. neoformans* survival by stabilizing cytoskeletal components and/or trafficking resistance factors to the cell surface.

*C. neoformans* has also evolved proteomics responses to survive within the phagosome, contributing to its ability to evade killing by the immune system and cause immune dysregulation. Following phagocytosis by innate immune cells, phagosomes containing ingested fungal cells are fused with the acidic lysosome to kill the pathogen with degradative compounds, reactive oxygen species, and nitric oxides [93]. Missall et al. conducted proteomics analyses of *C. neoformans* grown in host-like oxidative and nitrosative conditions and identified stress-induced Cryptococcus proteins that enhanced survival [94,95]. They revealed that thioredoxin-dependent thiol peroxidase (Tsa1) and transaldolase (Tal1) were important for both oxidative and nitrosative stress response, and that the glutathione reductase (Glr1) was important for only the nitrosative stress response, showing how *C. neoformans* employs diverse response strategies to survive host conditions.

### 3.4. Ubiquitin–Proteasome Pathway

As described previously, Geddes et al. found that PKA signaling enhanced the production of proteins involved in the ubiquitin–proteasome pathway (UPP) [26]. Other groups have proteomically identified proteins associated with the UPP that implicate it in boosting *C. neoformans* survival in adverse conditions. The F-box proteins comprise part of the SCF E3 ligase complex and are required for targeted ubiquitination of molecules [96]. Liu et al. characterized the role of *C. neoformans* F-box proteins and found that Fbp1 was critical for permitting cryptococcal infection and dissemination, and that deleting Fbp1 impaired virulence as well as sporulation and reproduction [97]. Liu et al. went on to use proteomics to elucidate the mechanisms of Fbp1-mediated survival by identifying the substrates that Fbp1 interacts with [98,99]. In their 2014 study, they identified that Fbp1 facilitated intracellular survival and growth within macrophages and that this was caused by Fbp1 association with the inositol phosphosphingolipid-phospholipase C1 (Isc1). Moreover, Han et al. used the labeled iTRAQ proteomics method to identify zinc-binding protein Zbp1 as another associated protein to Fbp1 [99]. Deletion and overexpression of Isc1 and Zbp1 both negatively impacted *C. neoformans* survival, pointing towards a regulatory role of Fbp1 in balancing target protein expression to optimize adaptations in adverse conditions.

Ubiquitination of proteins by ubiquitin ligases is dependent on adaptor proteins that facilitate binding of the ligase to the target protein [100]. The molecular dynamics and targets of *C. neoformans* ubiquitination were analyzed by employing proteomics analysis of the ubiquitin ligase Rsp5 and the arrestin adaptor proteins [101,102]. Four α-arrestins were identified through analysis of the *C. neoformans* proteome by Telzrow et al. in 2019: Ali1, Ali2, Ali3, and Ali4 [101]. Through protein-association proteomics, they showed how arrestin proteins facilitated the trafficking of fatty acid synthases Fas1 and Fas2 to the cell surface and promoted virulence. Du Plooy et al. subsequently used a proteomics method focused on ubiquitinated proteins to show that these arrestins, especially Ali2, facilitated ubiquitination of virulence-related chitin synthase proteins Chs4 and Chs5 by complexing with Rsp5 [102]. Deletion of Rsp5 led to severe cell surface defects that compromised *C. neoformans*’s ability to tolerate high temperatures.

Indeed, these studies support the notion that proteostasis or inhibiting the UPP is a viable method for impairing cryptococcal infection and dissemination, and that proteomics can be used to identify targets of ubiquitination.

### 3.5. Spores and Biofilms

The understudied spore and biofilm phenotypes are important to investigate as these phenotypic states increase the modes by which *C. neoformans* can enter the body, either as a dormant spore or on the surface of a medical device. Proteomics has helped reveal the mechanisms that enable *C. neoformans* to produce these phenotypes.

A survival strategy utilized by Cryptococcus is the ability to sporulate or form metabolically inactive spores that germinate when environmental conditions become favorable [103,104]. Sporulation is the main method by which fungi spread progeny throughout the environment [105], and may play a role in how cryptococcal infections initiate. Huang et al. utilized high-sensitivity mass spectrometry-based proteomics and found that *C. deneoformans* spores are enriched in protein pathways involving replication and chromosome biology, transcription and splicing, cellular transport, and carbohydrate metabolism [103]. They also found a protein of uncharacterized function, Isp2, that influenced the rate of sporulation and germination. In total, Huang et al. reported on 18 enriched proteins in the *C. deneoformans* spore phenotype that played various roles in sexual differentiation, spore formation, and germination. Future studies should explore the proteomic profile of *C. neoformans* spores to determine whether there is overlap in key proteins involved in sporulation and germination.

*C. neoformans* has demonstrated an ability to produce biofilms from shed capsule components and colonize medical implants such as prosthetic dialysis fistulae and ventriculoarterial shunt catheters [106,107,108,109]. Biofilm-associated *C. neoformans* is significantly more resistant to antifungals such as amphoteric B and caspofungin, placing patients with already elevated disease risk factors at greater danger if their medical implants are colonized with antifungal-resistant *C. neoformans* biofilms [110]. To identify the pathways involved in biofilm generation and maintenance, Santi et al. used shotgun proteomics to compare the proteomes of biofilm and planktonic (non-biofilm) *C. neoformans* cells. They identified 76 enriched proteins in the biofilm state, 33 of which were previously uncharacterized. Their analysis also showed that *C. neoformans* biofilms increased production of oxidative stress-response proteins including heat shock protein 70, a member of the Hsp90 family of proteins, increased extracellular proteolytic proteins including elastinolytic metalloproteinase, decreased protein translation, and changed metabolic states. In a subsequent study, Santi et al. expanded their analysis and compared the biofilm proteomes of *C. neoformans* to *Cryptococcus gattii*, another Cryptococcus pathogen that is implicated in approximately 5% of CM disease [111,112] (Table 1). Interestingly, they found that *C. neoformans* and *C. gattii* biofilms differed in the production of proteins related to the electron transport chain, with *C. neoformans* biofilms downregulating them while *C. gattii* biofilms upregulate them. Overall, the use of proteomics analysis has helped researchers advance our understanding of unique *C. neoformans* phenotypes like spores and biofilms to identify proteins and pathways that control their initiation and maintenance.

### 3.6. Acteylation

The benefit of proteomics analysis is the ability to detect changes that may otherwise be missed in conventional phenotype assays. Sirtuins are NAD+ histone deacetylases that modify histone acetylation to tighten around chromatin and silence transcription of genes in those regions [113]. In their study of *C. neoformans* sirtuins, Arras et al. used Sequential Window Acquisition of all Theoretical Mass Spectra (SWATH-MS) proteomics to determine whether their deletion strains were showing proteomic differences that were not detected by growth on phenotype-inducing agar plates [114]. They found that despite unclear results from growth assays, all five sirtuin deletion strains displayed perturbations in metabolic pathways such as carbohydrate metabolism, nitrogen metabolism, central carbon metabolism, and protein biosynthesis. The role of fungal acetylation in *C. neoformans* was further evaluated by Li et al. who analyzed the acetylome to identify important acetylation-dependent virulence proteins [46]. They reported that protein lysine acetylation (Kac) was important for regulating virulence and tissue colonization and that histone deacetylase (HDACs) deletion may be more effective at impairing *C. neoformans* virulence than sirtuins.

### 3.7. Other Discoveries

The proteomics analysis of *C. neoformans* has led to the discovery of previously unknown proteins, pathways, and peptides that play important roles in *C. neoformans* survival and virulence. Indeed, proteomics has revealed roles for short open reading frame-encoded peptides (sPEPs) [60], the pH regulating V-ATPase RAVE complex [115], casein kinase signaling pathways [116], and sulfur amino acid biosynthesis in supporting fungal survival in host-like environments [117]. These studies have demonstrated the utility of proteomics tools for elucidating the various mechanisms of *C. neoformans* survival strategies and identifying candidate targets for new antifungal treatments.

## 4. Proteomics of Cryptococcosis

The use of mass spectrometry-based proteomics to investigate host–pathogen interactions has expanded our understanding of host immune response strategies, Cryptococcus virulence mechanisms, and novel therapeutic modalities. In this section, we will discuss the findings of proteomics studies on cryptococcosis and antifungal treatment that have informed the next generation of clinical strategies for cryptococcosis.

### 4.1. Human Cryptococcosis

*C. neoformans* causes CM despite the initial cryptococcal infection beginning as a subclinical or latent lung infection [11,118]. While it is well established that immunosuppression is the primary risk factor that promotes fungal dissemination from the lungs, it is poorly understood how cryptococcal infection turns into systemic disease.

Many individuals who develop CM are co-infected with HIV, which depletes protective CD4+ T cells and increases the likelihood of fungal escape from the lungs [3,4]. To better understand how HIV and CM affect the brain, Selvan et al. investigated the proteomics profiles of frontal lobes in individuals with HIV and CM using iTRAQ proteomics to identify enriched proteins involved in disease pathogenesis [119]. They identified 317 differentially expressed proteins with various functions and roles including in morphogenesis, adhesion, immune system processes, and endocytosis. In contrast to singular viral infection where human leukocyte antigen (HLA) expression is repressed, CM/HIV co-infection significantly increased the expression of HLAs with immunohistochemistry (IHC) localization identifying invading histiocytes as the primary HLA-positive targets. Selvan et al. also identified a significant reduction in myelin proteins, which cover and protect neurons, in CM/HIV co-infected brains, which was confirmed by IHC staining. Proteins known to be involved in facilitating *C. neoformans* invasion were identified including the intracellular adhesion molecule Icam1 and the caveolin protein Cav1. Lastly, proteins of unknown contributions to cryptococcal disease were found such as the coagulation cascade protein fibrinogen beta chain Fgb and the astrocyte heat shock protein Cryab. This study was the first of its kind to use proteomics approaches to study the brains of patients infected with HIV presenting with CM and identify the various protein pathways and proteins involved in the disease manifestation.

While brain infection remains the most common cause of cryptococcosis-related mortality, most if not all cryptococcal infections begin in the lungs. Future studies should examine the proteomics profiles of human lungs with latent, subclinical, and/or symptomatic cryptococcal infection to identify proteins and pathways related to *C. neoformans* pathogenesis that play roles in fungal dissemination.

### 4.2. Infection Modeling

Cryptococcal infection modeling is critically important for understanding the complex host–pathogen interactions that determine fungal clearance or survival. Specific *C. neoformans* responses and biological phenomena can be studied by selecting models that induce the phenotype or interactions of interest. In this section, we will discuss the findings of proteomics studies of different models of cryptococcal infection.

*C. neoformans* is primarily killed or contained by the activity of innate phagocytes such as alveolar macrophages and astrocytes (macrophages of the CNS). Phagocytes engulf and ingest their targets into phagosomes which fuse with acidic lysosomes that contain cytotoxic and degradative compounds to kill and break down materials. *C. neoformans*, however, is well suited to avoid killing by macrophages through various virulence factors that help evade detection, block phagocytosis, neutralize lysosomal compounds, and/or escape the phagosome (vomocytosis) [24,120,121,122]. Studies using in vitro macrophage–*C. neoformans* interactions were used to identify proteomic shifts that occur in macrophages infected with *C. neoformans* [47,123]. Sukumaran et al. used quantitative proteomics to study *C. neoformans*- and *Klebsiella pneumoniae*-infected murine macrophages and identified host immune responses to fungal, bacterial, and co-infection [123]. They found that upon initial cryptococcal infection (90 min), macrophages differentially express 307 proteins involved in DNA replication, immunoglobulin binding, tRNA and rRNA binding, and viral responses, while macrophages infected for longer (48 h) exhibited decreases in immune response proteins. This suggests that macrophages become tolerant of *C. neoformans* over time, which may impair their fungicidal capabilities. Moreover, Sukumaran et al. showed that *C. neoformans* increased production of proteins with roles in biosynthesis catabolism and metabolism during initial infection but only catabolism and cellular regulation after 48 h. These findings reveal how macrophage responses to cryptococcal infection change over time, and how *C. neoformans*’s survival can cause macrophages to become tolerant and ineffective at clearing them.

The surviving *C. neoformans* cells within phagocytes are not only able to survive but also use the host cells as reservoirs to replicate [124,125]. This ability of *C. neoformans* to hijack phagocytes and thrive within them is believed to contribute to its capability to escape from the lungs and disseminate to other organs, described as the Trojan horse theory of dissemination [125,126]. Inhibiting intracellular fungal replication is essential for preventing *C. neoformans* infections from using host cells to escape tissue confinement and enter critical organs. Using phosphoproteomics, Pandey et al. discovered that signaling between the AMPK pathway and autophagy initiation complex (AIC) promoted fungal phagocytosis and replication [55]. Cryptococcal infection caused active phosphorylation of proteins in the AIC, which localized to phagosomes containing fungal cells and activated the AMPK pathway. By inhibiting AMPK using compound C (CC), phagocytosis and autophagy were reduced, causing a significant reduction in fungal replication. Moreover, when mice infected with *C. neoformans* were treated with CC, they displayed lower fungal burden despite no changes in immune cell counts and cytokines. These data highlight how important intracellular replication is for *C. neoformans*’s survival, and that blocking this immune evasive strategy is an effective method for disrupting *C. neoformans*’s ability to sustain an infection.

*C. neoformans*-infected macrophages release extracellular vesicles (EVs) that alter cell signaling pathways and protein production in nearby cells. Macrophage activity has been shown to be mediated by EVs and influence pathogen response [127]. Zhang et al. used proteomics to analyze the changes in human macrophages treated with EVs secreted from *C. neoformans*-infected macrophages [128] (CnIMs) and found that CnIM EVs were highly pro-inflammatory. They showed that CnIM EVs contained immune cell-recruiting alarmin proteins, heat shock protein Hspa9, and the glutamatergic signaling regulator Gria1, highlighting their role in priming cells to respond to an infection. Recepient macrophages were activated into a pro-inflammatory phenotype and had significantly elevated production of follistatin-like 3 (Fstl3), which is associated with inflammation. They reported that treating *C. neoformans*-infected mice with CnIM EVs reduced fungal burden, but increased mortality. Indeed, this is in agreement with clinical studies showing that administering pro-inflammatory exogenous interferon γ did not improve survival, despite decreases in fungal burden [23]. These findings are informative as they reveal how phagocytes communicate with one another during cryptococcal infection to enhance fungicidal capabilities and how host responses require a careful balance across immune mechanisms to avoid death.

Muselius et al. investigated the temporal splenic proteome using a dual-perspective approach to identify biomarkers of disseminated fungal disease [6]. They found that by 3 days post-infection, the splenic proteome began changing in response to fungal proteins without detection of *C. neoformans* cells. They attributed the source of these fungal proteins to antigen-presenting cells (APCs) from the lungs initiating the adaptive immune response. Fungal pattern-recognizing lectin-like receptors increased in response to fungal antigen presentation by 21 days post-infection. Known *C. neoformans* virulence factors Pkr1, CipC, urease, and α-amylase were detected in the spleen, possibly representing the APC-presented antigens. These results are important as they implicate how specific proteins from cryptococcal infection are trafficked to lymphocyte-activating lymphoid organs early on and show the resulting changes in the splenic proteome.

## 5. Evaluating Novel Clinical Interventions

### 5.1. Antifungals and Vaccines

The current standard of care is insufficient for addressing the global burden of cryptococcosis and CM. To advance the development of new treatments, it is imperative that candidate antifungals be fully evaluated for their fungicidal or fungistatic capabilities and their possible off-target effects. Proteomics tools are well suited for studying candidate antifungals as they facilitate the analysis of both fungal control mechanisms and off-target host effects. Here, we discuss some of the findings from studies using proteomics to investigate novel antifungal compounds.

Benzimidazoles are a class of antiparasitic compounds that have recently been shown to be effective against *C. neoformans*, with fenbendazole showing particular value [129,130,131]. Fenbendazole works by disrupting microtubule organization in both parasites and *C. neoformans* and is especially effective against pathogens ingested by macrophages [131]. To determine the global cellular response, de Oliveira et al. analyzed the proteomic profile of *C. neoformans* treated with fenbendazole [132]. They found that fenbendazole significantly disrupted the production of 200 proteins involved in major cellular functions including RNA processing, intracellular traffic, and DNA replication. These impacted pathways are heavily dependent on proper microtubule function which supports the hypothesis that fenbendazole’s primary mechanism of action is microtubule disruption. They also reported that protein kinases Chk4, Tco2, Tco3, Sch9, and Bub1 were required for fenbendazole activity, with mutations in these genes causing phenotypic disturbances similar to fenbendazole treatment. These findings unveil fenbendazole’s global effects in *C. neoformans* and provide evidence that supports its role as an antifungal.

As the story of penicillin reminds us, there exists a wide selection of naturally occurring antimicrobials with targeted effects that are good candidates for novel therapies. Gutierrez-Gongora et al. found that extracts from mollusks and mussels are able to impair *C. neoformans* growth, survival, and virulence factor production [133,134]. In their 2023 study, Gutierrez-Gongora et al. investigated crude and clarified extracts from three mollusk species, *Planorbella pilsbryi*, *Cipangopaludina chinensis*, and *Cepaea nemoralis*, and determined the *C. neoformans* pathways impacted by treatment [133]. Their main findings were that extracts from each mollusk worked differently in impairing various phenotypic characteristics which were validated by their proteomic profiles. *P. pilsbryi* extracts impaired heat stress responses and fungal growth, with the proteomic profile reflecting disruptions in lipid and actin binding. *C. chinensis* extracts inhibited biofilm formation and caused defects in ATP binding and cytoskeleton interactions. *C. nemoralis* disrupted capsule production and proteins involved in ATP binding and β-hexosaminidases. In 2024, Gutierrez-Gongora et al. examined extracts from two freshwater mussels, *Dreissena polymorpha* and *Lasmigona costata*, for antifungal activity. Extracts from both mussels negatively impacted *C. neoformans* heat tolerance and biofilm production and enhanced fluconazole sensitivity, while extracts from *D. polymorpha* also disrupted capsule production. Using bottom-up proteomics, they identified five extracts as being calcineurin-like proteins that may impair heat tolerance by competitively inhibiting *C. neoformans* Ca^2+^ sensing. They also determined that the inhibition of cysteine peptidases and the Rim pathway caused by *D. polymorpha* extracts contributes to capsule dysfunction. The findings of these studies not only showcase the potential of these extracts as candidate antifungals but also relate disruptions in key *C. neoformans* virulence factors with specific proteomic perturbations that inform the synthesis of new compounds.

Perhaps one of the most effective methods of treating CM is to prevent its occurrence in the first place. Current dogma attempts to accomplish this by screening at-risk patients for signs of disseminated cryptococcal disease, such as with CrAg LFA strips or blood testing [11,12]. However, these methods only facilitate earlier detection of disseminated disease and may not catch the infection before the brain is colonized [14,15]. Moreover, individuals who do not have known risk factors, are in resource-limited settings that preclude preventative screening, or have fungal burdens below limits of detection may still develop cryptococcal disease that leads to CM. Instead, immunization is an alternative strategy that can be used to enhance the body’s natural capability to control infections and could be applied universally [135,136,137]. Here, we discuss some of the advances in anticryptococcal immunization strategies discovered from proteomics analysis.

*C. neoformans* EVs contain numerous immunogenic virulence factors and fungal components and possess the ability to prime the adaptive immune response which has led to them being investigated as possible vaccine candidates. To determine possible candidate immunogens within the *C. neoformans* EV, Rizzo et al. utilized proteomics to identify shared proteins between three *Cryptococcus* spp. EVs: *C. neoformans*, *C. deneoformans*, and *C. deuterogattii* [32]. They identified 71 EV-associated proteins in total and found that 17 proteins were shared amongst all tested *Cryptococcus* spp. including the major virulence factor chitin deacetylase and enzymes involved in polysaccharide degradation and modification. Other EV-associated proteins had roles in metabolic pathways in copper scavenging and the ETC. To test whether EVs would confer protection against infection, Rizzo et al. generated EVs from an acapsular *C. neoformans* strain (cap59Δ) and treated mice with intraperitoneal injections prior to intranasal cryptococcal infection. The cap59Δ EV immunization successfully prolonged survival in mice. These data underscore the benefit of using proteomics to identify immunogenic proteins from complex vesicular compartments that can translate to vaccine development.

Selecting strong immunogenic epitopes that facilitate protection against several strains and species of Cryptococcus is key to developing an effective anticryptococcal vaccine. However, identifying epitopes from basic science experiments can be time-consuming and cost-prohibitive. Informatics tools can bridge this gap to accelerate the identification of candidate epitopes by bioinformatically scavenging the organism’s proteome for particular motifs and structural components [138]. Zhu et al. utilized subtractive proteomics and identified four *C. neoformans* epitopes with predicted coverage of serotype A and D *C. neoformans* strains: one glucosidase (J9W2B9) and three β-glucan synthesis-associated proteins (J9VJK0, J9VLU5, and J9VRH0) [51]. Two theoretical vaccines were developed, CAV1 and CAV2, which were computationally evaluated through simulations on vaccine–receptor molecular docking, binding energy dynamics, and immune stimulation and coverage. The results of the simulated immunization strategy predicted strong pro-inflammatory induction and a global coverage of 98.68%. While these predictions need to be tested in biological models, the ability to use proteomic data to develop theoretical vaccines with predicted dosing strategies and immune responses is a powerful tool for accelerating vaccine discovery.

### 5.2. Clinical Identification Using Mass Spectrometry

One of the greatest clinical applications for mass spectrometry-based techniques is its use in identifying unknown organisms in samples based on protein expression or proteomic profiles [139,140]. This is especially useful for the clinical diagnosis of CM due to it being primarily caused by two Cryptococcus species: *C. neoformans* and *C. gattii*. Differentiation of these *Cryptococcus* spp. is difficult with routine laboratory methods [71,141]. Matrix-Assisted Laser Desorption/Ionization Time-of-Flight (MALDI-TOF) is a method that utilizes mass spectrometry to identify organisms based on the spectra of proteins, peptides, or intact cells that has become increasingly used to differentiate Cryptococcus and fungal infections [140,142,143,144,145,146].

The utility of MALDI-TOF for Cryptococcus sp. identification was shown by Firacative et al. and McTaggart et al. who used it to correctly identify 100% of *C. neoformans*, *C. gattii*, and hybrid strains [144,145]. Firacative et al. generated 1120 mass spectra to create a spectral library for each of the eight major molecular types that divide the Cryptococcus sp. complex [144]. McTaggart et al. similarly generated spectral libraries to identify 160 yeast isolates [145]. The major molecular types are useful genotyping classifiers that distinguish strains geographically and epidemiologically [147,148,149]. Thus, the findings of these studies show how mass spectrometry can provide useful information on *C. neoformans*’s geographical source and possible pathogenicity.

In time, as more research into the proteomics of Cryptococcus strains unveils profiles associated with worse disease outcomes, we expect that MALDI-TOF identification techniques will be replaced with more specific point-of-care proteomics analyses to stratify patients for targeted care.

## 6. Future Directions

The use of proteomics to study *C. neoformans* and cryptococcal infection has deepened our understanding of host–Cryptococcus interactions, pathways involved in virulence factor activity, immunogenic antigens, fungal survival, therapeutic responses, and host defense mechanisms. However, limitations in proteomics studies include the lack of standardized proteomic workflows, difficulty in isolating Cryptococcus peptides from capsule and host tissue samples, incomplete *C. neoformans* references and annotations, and the prohibitive cost of conducting proteomics.

The Cryptococcus capsule poses a unique challenge in proteomics sample preparation due to the abundance of hydrophobic and lipophilic components. Sample preparation techniques often discard fractions containing capsule components, leading to the loss of immunogenic capsule-embedded proteins and antigens. Moreover, simultaneous analysis of host and *C. neoformans* proteins from infected tissue impairs fungal protein detection due to the substantially higher abundance of host proteins. These issues can be minimized by establishing standardized sample preparation workflows and improving the detection of less-abundant proteins. Lastly, despite the large diversity in Cryptococcus sub-species used in proteomic research, UniProt possesses a reference proteome for only *C. deneoformans*. To establish additional reference proteomes that better align with the various Cryptococcus models, other proteomes need to be reviewed and annotated for improved accuracy. These limitations pose barriers against the widespread adoption of mass spectrometry-based proteomics and should be addressed to increase the adoption of proteomics techniques in *C. neoformans* research.

To bridge the gap between the bench and bedside, more proteomic studies need to be conducted using clinically relevant *C. neoformans* strains and models. Indeed, the use of patient-derived isolates has greatly benefited the field by providing infections that more closely mimic those seen in human disease [4,18,22,146,150,151]. As shown by Jackson et al. in their 2024 *Nature Communications* publication, clinical isolates between patients differ greatly in disease manifestation and single-nucleotide polymorphisms even if the infection occurred in a similar region [151]. It is clinically relevant, therefore, to quickly and accurately determine the virulence of an infecting strain to stratify patients based on predicted outcomes and enable targeted care. As mass spectrometry continues to become more utilized in clinical settings, mass spectrometry-based proteomics can be a powerful tool that bridges the gap between modern treatments and more effective individualized therapy.

## 7. Conclusions

*C. neoformans* morbidity and mortality remain a serious problem globally and cause approximately 19% of AIDS-related deaths [3]. While advances in fungal dissemination detection have improved early diagnosis, seronegative CM, high mortality rates, and antifungal resistance prove that more work is needed to address cryptococcal disease. Mass spectrometry-based proteomics is an increasingly effective technique for studying protein interactions and pathways involved in cryptococcosis. Advances in proteomics have revealed novel insights into virulence factors, stress response pathways, infection models, host responses, and treatment efficacy, which have informed the next generation of *C. neoformans* research. The field will benefit from further advancement in standardized sample preparation, mass spectrometry hardware, data analysis, and databases specific to the complex biology of *C. neoformans*. By continuing to improve *C. neoformans* proteomic research, we will discover novel targets for clinical intervention and develop better antifungal therapies to benefit patient outcomes.

## Figures and Tables

**Figure 1 jof-11-00529-f001:**
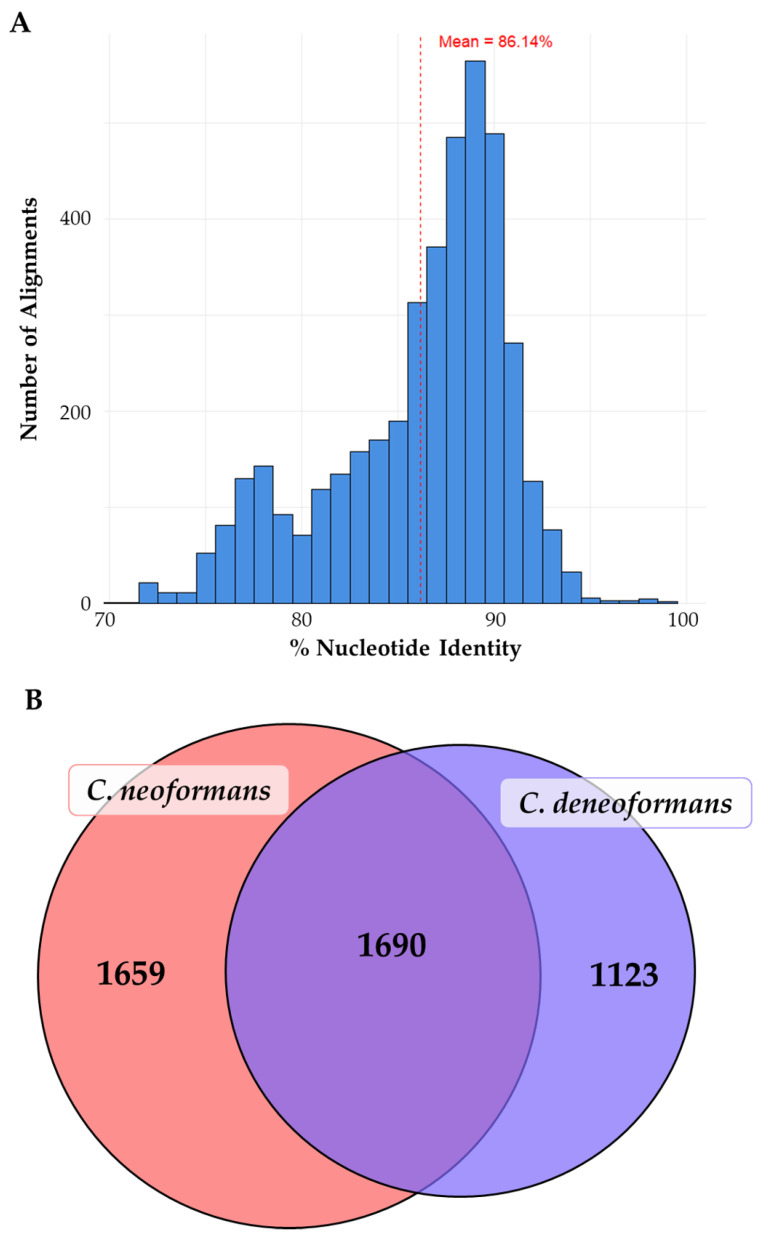
*C. neoformans* and *C. deneoformans* reference proteomes encode different proteins despite being genetically similar. (**A**) Genome alignment of *C. neoformans* (CN3) and *C. deneoformans* (ASM9104v1) genomes was performed using BLAST+ to visualize the distribution of percent nucleotide identity for each alignment. The mean percent nucleotide identity (86.14%) was calculated to determine the overall relatedness between genomes. (**B**) Protein identification from peptides isolated from a serotype A *C. neoformans* clinical isolate using the *C. neoformans* (pink, Proteome ID UP000010091) and *C. deneoformans* (purple, Proteome ID UP000002149) proteomes was performed using DIA-NN to determine the overlap of protein detections. The *C. neoformans* proteome search detected 3349 proteins and the *C. deneoformans* search detected 2813 proteins with 1690 proteins (37.7%) shared between both searches.

**Table 1 jof-11-00529-t001:** Cryptococcus strains used in proteomics research.

Strain	Cryptococcus Species	Serotype	Origin	Characteristics	UniProt Proteome ID/Number of Proteins
H99	*C. neoformans*	A	Isolated in 1978 from an American patient with Hodgkin’s Lymphoma	Representative of environmentally derived *C. neoformans*, mating type locus α	UP000010091/7429
KN99α	*C. neoformans*	A	Laboratory-generated strain derived from H99	Virulent in in vivo models and reproduce robustly, mating type locus α	UP000232048/7422
JEC21	*C. deneoformans*	D	Laboratory-generated strain	Representative of *C. deneoformans*, mating type locus α	UP000002149/6740
R265	*C. gattii/C. deuterogattii*	B	Isolated from 1999 Vancouver Island Outbreak	Virulent in immunocompetent individuals, mating type locus α	UP000029445/2945

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
