# Peer review of "Insights from Mass Spectrometry-Based Proteomics on Cryptococcus neoformans"

_jof, 2025, doi:10.3390/jof11070529_

Round 1
Reviewer 1 Report
This is an exceptionally well written and timely review on proteomic approaches to better understand the biology of Cryptococcus species, especially in light of their pathogenesis. The review is comprehensive, and it presents the relevant data in a balanced and fair manner. It will provide an exceedingly important resource for the scientific community to understand what has currently been accomplished in the way of proteomics of Cryptococcus, and it directs future important new investigations.
There is perhaps not a clear answer to this, but the authors may want to comment further on the data presented in Figure 1B. It is not clear from their description if they propose that the proteomes of these two related species are truly as different as measured? Could this be an artifact of peptide identification, or differences in protein expression under theses conditions? Could any of these proteome differences be explained by the small differences in the genome?
Line 359. Lower case g in "gattii" and mispelled "Cryptococcus"
Author Response
We thank Reviewer 1 for their constructive edits and suggestions. Here are our responses:
Comment: Line 359. Lower case g in "gattii" and misspelled "Cryptococcus"
Response: Correction implemented.
Comment: There is perhaps not a clear answer to this, but the authors may want to comment further on the data presented in Figure 1B. It is not clear from their description if they propose that the proteomes of these two related species are truly as different as measured? Could this be an artifact of peptide identification, or differences in protein expression under theses conditions? Could any of these proteome differences be explained by the small differences in the genome?
Response: While it is still unclear why there are differences in protein identifications between the JEC21 and H99 proteome searches in Fig. 1B, it is partially due to differences in proteome entries on UniProt. The JEC21 entry has 6740 proteins while the H99 proteome contains 7429 demonstrating 699 proteins that are only in the H99 proteome. However, this accounts for ~9% of the differences in protein identification while we showed 37.7% in our analysis. Proteome entries in UniProt are predicted from submitted genomes and then annotated afterwards to confirm the protein entries. The JEC21 proteome is defined as the C. neoformans “reference proteome” on UniProt and is better annotated than the other proteomes. Therefore, we speculate these differences are due to a combination of genetic and proteomic differences, annotation quality, and other undetermined confounding factors. Ultimately, any unexplained confounding factors will need to be determined through focused investigation.
For clarity, we rephrased the final sentence to “Only 1690 proteins (37.7%) overlapped between both searches. Importantly, proteome entries in UniProt are predicted from submitted genomes and then annotated afterwards to confirm the protein entries. The JEC21 proteome is defined as the C. neoformans “reference proteome” on UniProt and is better annotated than the other proteomes. Thus, the observed differences between the two Cryptococcus proteomes likely underscores the importance of creating high-quality and well-annotated reference proteomes for the various Cryptococcus species.”.
Reviewer 2 Report
Betancourt and Nielsen prepared a fluent, critical​, interesting, and well written review on the knowledge achieved ​by the numerous proteomics works reported on Cryptococcus neoformans. The authors travel through the many fungal virulence factors already described and molecules involved in host-fungal interaction, showing how proteomics helped unveil their role in pathogenesis and how they can still be used to select molecules to be used in diagnosis, vaccination and treatment.
l. 42-44: are the numbers updated for 2025 or do they refer to reference 3, with data from 2020. Please specify in the text.
lines 64-67: please rephrase to make it clearer.
line 181: consider using "expression" instead of "production" here and in other parts of the text, as well as "express" instead of "produce".
lines 336, 340: please cite the numerical reference after Huang et al. In line 340, please delete the comma after et al.
line 366: please correct typo
Author Response
We thank Reviewer 2 for their constructive edits and suggestions. Here are our responses:
Comment: l. 42-44: are the numbers updated for 2025 or do they refer to reference 3, with data from 2020. Please specify in the text.
Response: We added “in 2020 was” in L.42.
Comment: lines 64-67: please rephrase to make it clearer.
Response: We rephrased to “However, clinical trials investigating the use of adjunctive interferon γ, the prototypical type-1 cytokine, to treat HIV-associated CM showed no benefit on mortality rates despite reductions in fungal burden [23].”.
Comment: line 181: consider using "expression" instead of "production" here and in other parts of the text, as well as "express" instead of "produce".
Response: While the term “expression” can be used to describe the increased or decreased presence of protein in a sample, proteomic approaches generally cannot determine whether the increased or decreased presence of protein is due to transcriptional regulation (i.e. expression) or differences in degradation, localization, splicing, or post-translational modifications unless specifically tested. We chose to use the word “produce” over “expression” to minimize confusion over the mechanisms that ultimately lead to the increased or decreased identification of proteins in a sample.
Comment: lines 336, 340: please cite the numerical reference after Huang et al. In line 340, please delete the comma after et al.
Response: We added the correct citation.
Comment: line 366: please correct typo
Response: Typo was corrected.
Reviewer 3 Report
This is a well-written and comprehensive review that provides an insightful and timely synthesis of mass spectrometry-based proteomics applied to Cryptococcus neoformans biology and pathogenesis. The authors have done an excellent job summarizing a wide array of studies that span virulence factors, host-pathogen interactions, stress response pathways, and even translational applications such as diagnostics and vaccine development. The structure is logical, and the narrative is clear, making it a valuable resource for researchers in medical mycology, proteomics, and infectious diseases.
The depth of literature coverage and critical integration of findings make this review especially strong. I appreciate the emphasis on how proteomics informs both basic and translational research. Furthermore, the inclusion of future directions and discussion of technical challenges adds thoughtful context and encourages ongoing exploration in this area.
That said, I suggest a minor revision to further improve clarity and utility for readers:
-
Figure/Table Suggestions: While the manuscript includes one figure (Figure 1), additional schematic diagrams or summary tables would help consolidate complex information—for example, a table summarizing major proteomic discoveries across different virulence factors or stress conditions.
-
Strain Clarification: The manuscript refers to various Cryptococcus strains and serotypes (H99, KN99, JEC21, etc.). It would benefit from a brief table or figure summarizing these strains, their relevance, and common uses in proteomics studies to aid readers unfamiliar with the nuances.
-
More Critical Perspective on Gaps: While the review is rich in coverage, a more critical reflection on the limitations of current proteomic studies, such as host contamination in samples or reproducibility across labs, would strengthen the discussion section.
-
Minor Language and Formatting Edits: There are a few minor typographical and grammatical issues (e.g., in Section 3.6 "Acteylation" should be "Acetylation"). A careful proofreading pass will help polish the manuscript.
Overall, this is a strong and informative review that will be a valuable contribution to the field. I recommend minor revision prior to acceptance.
Please see Major comments
Author Response
We thank Reviewer 3 for their constructive edits and suggestions. Here are our responses:
Comment: Figure/Table Suggestions: While the manuscript includes one figure (Figure 1), additional schematic diagrams or summary tables would help consolidate complex information—for example, a table summarizing major proteomic discoveries across different virulence factors or stress conditions.
Response: While we understand the desire to consolidate the information in this review, we believe a summary graphic or table depicting the major proteomic discoveries across the various experimental conditions would be misleading and would fail to adequately consolidate the complexity across the experimental findings. Most of the studies used differing methodologies such as protein interaction networking, entire organism phosphorylation and secretion profiling, in vitro immune cell models, host-pathogen interaction studies, and clinical responses. Many of these differences would not be adequately addressed in a summary figure and readers could misinterpret the underlying critical differences between the studies. Instead, we believe our compartmentalization of the studies under informative sections and sub-headers still summarizes the findings while allowing important distinctions to remain transparent.
Comment: Strain Clarification: The manuscript refers to various Cryptococcus strains and serotypes (H99, KN99, JEC21, etc.). It would benefit from a brief table or figure summarizing these strains, their relevance, and common uses in proteomics studies to aid readers unfamiliar with the nuances.
Response: Having a resource to help readers understand the various Cryptococcus organisms is useful, especially given the numerous nomenclature changes that have occurred. We included this as Table 1 and included the UniProt proteome ID and number of proteins for each of the reference strains in the Cryptococcus species complex
Comment: More Critical Perspective on Gaps: While the review is rich in coverage, a more critical reflection on the limitations of current proteomic studies, such as host contamination in samples or reproducibility across labs, would strengthen the discussion section.
Response: Greater detail on the current limitations of proteomic studies was added to Section 6. Future Directions.
Comment: Minor Language and Formatting Edits: There are a few minor typographical and grammatical issues (e.g., in Section 3.6 "Acteylation" should be "Acetylation"). A careful proofreading pass will help polish the manuscript.
Response: Grammarly was used to check for grammatical errors, and relevant corrections were made. In addition, the manuscript was also manually corrected for typos.